# Xyloglucan–Cellulose Nanocrystals Mixtures: A Case Study of Nanocolloidal Hydrogels and Levers for Tuning Functional Properties

**DOI:** 10.3390/gels10050334

**Published:** 2024-05-15

**Authors:** Géraldine Rangel, Céline Moreau, Ana Villares, Christophe Chassenieux, Bernard Cathala

**Affiliations:** 1UR1268 BIA, INRAE, 44300 Nantes, France; geraldine.rangel@inrae.fr (G.R.); celine.moreau@inrae.fr (C.M.); ana.villares@inrae.fr (A.V.); 2IMMM UMR CNRS6283, Le Mans University Avenue Olivier Messiaen, 72085 Le Mans, France; christophe.chassenieux@univ-lemans.fr

**Keywords:** cellulose nanocrystals, xyloglucan, nanocolloidal hydrogels, mechanical properties, thermoresponsive, osmotic compression, self-healing

## Abstract

The development of fully biobased hydrogels obtained by simple routes and in the absence of toxic or environmentally harmful reagents is a major challenge in meeting new societal demands. In this work, we discuss the development of hydrogels made from cellulose nanocrystals (CNCs) and xyloglucan (XG), two non-toxic, renewable, and biobased components. We present three strategies to fine-tune the functional properties. The first one consists in varying the XG/CNC ratio that leads to the modulation of the mechanical properties of hydrogels as well as a better comprehension of the gel mechanism formation. The second relies on tuning the XG chains’ interaction by enzymatic modification to achieve thermoresponsive systems. Finally, the third one is based on the increase in the hydrogel solid content by osmotic concentration. The high-solid-content gels were found to have very high mechanical properties and self-healing properties that can be used for molding materials. Overall, these approaches are a case study of potential modifications and properties offered by biobased nanocolloidal hydrogels.

## 1. Introduction

Cellulose nanocrystals (CNCs) are biobased nanoparticles which, over the last two decades, have become both a subject of extremely active scientific research and an industrial reality. This remarkable development is due not only to their biobased and renewable nature, but also to the unique properties of these nanoparticles, such as their mechanical properties, their ability to establish numerous interactions (hydrogen, van der Waals, electrostatic) or to be chemically modified, and of course, their small dimensions and semicrystalline organization (i.e., their nanoparticle nature), which underpins their colloidal properties. All these characteristics have led the scientific community to use CNCs in an extremely wide range of applications, which are described in various reviews [1,2]. CNCs are increasingly used in blends with different compounds to adjust or enhance the properties of CNCs. This is particularly true for blends with polymers that can be ionic or neutral. In the frame of CNC/neutral polymer mixtures, the specific case of hemicellulose is particularly attractive because these entirely biobased mixtures can reproduce, at least partially, natural assemblies such as plant cell walls. These biomimetic mixtures therefore provide tools for increasing our knowledge, as well as opportunities for innovation. In this context, our group has been working for several years on mixtures of CNC and xyloglucan [3,4,5,6].

Xyloglucan (XG) is one of the hemicelluloses whose interaction with cellulose has been extensively studied and used for designing materials [7,8]. Its structure consists of a β-(1-4)-linked D-glucopyranosyl linear backbone with three α-(1-6)-linked xylose units, which can be substituted with β-(1-2)-linked galactosyl residues and in some cases, with arabinosyl or fucosyl residues as well [9]. Nanocellulose and xyloglucan mixtures therefore represent prime systems for gaining a better understanding of biobased nanoparticle/polymer mixtures and for developing innovative materials from renewable resources. Indeed, polymer–nanoparticle self-assembly is a simple route to control the dispersion of nanoparticles, rheological properties of the mixtures, and also to design tunable materials, such as hydrogels, without having to resort to complex synthetic chemistry approaches [10,11]. The latter belong to the recently reviewed nanocolloidal gels (NCGs) that are described as a class of soft-matter materials, in which nanoparticles (and nanoparticles/polymer complexes) act as building blocks of the colloidal network [12].

However, even if the mixing of a CNC dispersion and a polymer solution in water remains a simple operation compatible with the search for sustainable processes, the functionalities of the mixture depend on the balance between diverse and tunable interactions. The work presented in this article describes the interactions between XG and CNCs, based on XG adsorption on CNC surfaces followed by quartz crystal microbalance with dissipation (QCM-D) and discussed in the light of previous work in the literature. Then, the means of modifying the properties of the mixtures is discussed, with various results showing that it is possible to adjust the interactions at play within the mixtures. In a first part, the effect of the XG/CNC ratio is studied; in a second approach, the interaction between the XG chains is tuned by enzymatic modification; and finally, the XG/CNC concentration is increased to serve as a lever for the modification of the properties of the blend. Taken together, these elements make it possible to trace the behavior of XG- and CNC-based hydrogels and to propose a mechanistic vision of their structure/properties relationships. This systemic description makes it possible to examine various functional properties of hydrogels and consider possible applications. In addition, XG/CNC systems can be seen as functional building blocks in their own right (beyond XG and CNC) and also as model colloidal biobased nanogels made of hemicellulose and nanocellulose (or more generally of polymer and nanoparticles).

## 2. Results and Discussion

Mixtures of polymers and nanoparticles are complex composite systems, and their properties are not simply the sum of the properties of the individual partners as long as either attractive or repulsive interactions are at play [13,14,15,16,17]. The complexity of the interactions involved (intensity, range, ability to be modulated by the environment, etc.) and their intricacies offer an almost infinite range of possibilities for modulating the properties of nanocolloidal hydrogels. However, in the context of XG/CNC blends, three major levers can be identified as presented in Figure 1: (i) varying the polymer/particle ratio (XG/CNC weight ratio); (ii) varying the concentration (XG + CNC); and (iii) modifying the interactions between polymers [17]. These three modalities are generalizable to many systems, but the strategies and results presented in this paper are specific to the XG/CNC system, which can be considered as a model system for biobased nanocolloidal hydrogel [12,18].

### 2.1. Impact of the XG/CNC Ratio

The mixing of the CNC dispersion and XG solution is a simple operation which, under certain conditions, leads to the formation of a solid material, whereas the initial CNC dispersions and XG solutions behave as liquids. This change in physical state demonstrates unambiguously the existence of interactions (either attractive or repulsive) between CNC and XG during mixing. Interactions between XG and cellulose have been demonstrated and studied for several decades [7,8,19,20]. They have been highlighted using a variety of approaches but often within the framework of simplified model systems such as cellulose surfaces, which enable the various driving forces and relevant criteria to be identified so that more complex systems can then be apprehended [21,22]. Thus, in the case of XG/CNC mixtures, the adsorption of XG on a surface on which nanocrystals are deposited can be used as a model. The adsorption of molecules to a surface, whatever their size, can be divided into several phases, the first being diffusion to the surface, followed by contact (interaction). If the adsorption site is free, the next phase consists of polymer reorganization to maximize the number of interactions, leading to more extensive conformations (Figure 2c). If the adsorption site is already occupied, adsorption induces polymer reorganization, leading to the formation of loops and tails (Figure 2d). The final stages consist in minimizing free energy through a conformational reorganization according to the polymer surface concentration. In the case of polymers, all these steps depend both on the thermodynamics of the system (solvent–polymer interaction, polymer surfaces, etc.) and on the adsorption experimental parameters (polymer concentration, specific surface area, surface occupancy, temperature…), so kinetic information is crucial for assessing the final state of adsorption.

Quartz crystal microbalance with dissipation measurement (QCM−D) is an acoustic technique based on the piezoelectric properties of a quartz crystal. Initially, QCM-D was used to measure the frequency variation of the quartz crystal, which varies according to polymer adsorption (*Δf*). The frequency shift gives access to the quantity of adsorbed polymers, notably through the use of the relationship proposed by Sauerbrey [23]. However, in a liquid medium and in the case of a non-rigid layer, a more advanced exploitation of the technique was proposed by Rodahl et al. through the measurement of the damping of quartz oscillation, i.e., its dissipation factor *D* [24]. The adsorption of a non-rigid layer leads to a decrease in frequency and an increase in the dissipation factor (Figure 2a). By plotting the variation in dissipation (*ΔD*) as a function of frequency variation (*Δf*), we can distinguish between different dissipative behaviors and therefore different viscoelastic properties, depending on the interaction of the polymers with the surface and between polymers (Figure 2b). Rodahl et al. attributed the increase in *ΔD* to three internal-layer processes: (i) the deformation of a viscoelastic porous layer during oscillation; (ii) the displacement of liquid trapped in the pores; and (iii) the action of bulk liquid on the layer [24]. In the case of polymers, these three elements can be linked to the adsorption of polymers in a different conformation, i.e., a variation in the polymers’ fraction forming loops and tails related to the proportion of trains.

In the case of polymer adsorption, the ability of QCM-D to monitor polymer adsorption in real time enables us to study the evolution of the viscoelastic properties of the adsorbed layer. The adsorption of XG on a CNC surface was monitored by varying the concentration of the XG injected from very diluted solutions (0.3 μg mL^−1^) to more concentrated ones (15.0 μg mL^−1^), which is equivalent to varying the adsorption kinetics. The superimposed *ΔD* vs. *Δf* plots of the different injected solutions are shown in Figure 2e. Two domains, i.e., slopes, are clearly identifiable. At low frequency values (≤10 Hz), i.e., low concentrations of polymers, the slope is in the range of 0.07 and 0.1. At higher frequency variations (≥10 Hz), the slope increases to values of the order of 0.2. The lowest injection concentrations (0.3–2.5 μg/mL) show only one slope, whereas the two domains are present at higher concentrations (5–15 μg/mL).

Increasing the injection concentration therefore significantly modifies the viscoelastic properties of the adsorbed layer. When the concentration is low, the polymer defaults to the adsorption sites available on the surface and reorganizes its conformation to cover the maximum surface, leading to a high proportion of trains and a strong connection of nanocrystals (Figure 2c). On the contrary, at higher concentrations, the adsorption sites are rapidly occupied, and the arrival of excess polymer leads to the formation of loops and trains that limit the connection between the CNCs and draw in more water, leading to higher dissipation values for the same quantity of adsorbed polymer (Figure 2d). Since the CNC surface concentration remains constant, the increase in the XG injection concentration results in an increase in the XG/CNC ratio that is thus directly linked to the structure of the XG layer organization at the CNC surface. Therefore, the XG/CNC ratio seems to be a key factor dictating the properties of hydrogels since the structure of the XG surface layer influences the interactions between the CNC/XG complexes (physical cross-linking, steric repulsion, interaction between XG chains).

In order to assess the impact of the XG/CNC ratio, a phase diagram (XG/CNC/water) was established (Figure 3). XG/CNC mixtures were formulated for weight ratios ranging from zero to nine and concentrations from 2 to 20 g L^−1^. It should be noted that, in the concentration range studied, the XG and CNC solutions taken separately showed no gelation. Gel formation for the mixtures was assessed by the inverted-tube method, which involves formulating the gel in a test tube and then inverting it. Mixtures that can support their own weight do not flow and are therefore classified as gel, whereas mixtures that do flow are classified as liquid. Please note that for all samples, the properties were evaluated 5 min and 5 days after being prepared either visually or by rheology. No differences were observed, suggesting that the systems reached a steady state at least 5 min after being prepared. Figure 3 shows the sol/gel phase diagram relating the total CNC + XG concentration to the XG/CNC ratio.

Only the frontier conditions, i.e., the lowest gelification concentration for each ratio, have been plotted in Figure 3 to identify sol and gel domains. At low XG/CNC ratios, the gelling concentration decreased rapidly when increasing the XG/CNC ratio, reaching a minimum before increasing. Each frontier can be fitted by a parametric linear equation:(1)$$\mathrm{At}\;\mathrm{low}\;\mathrm{ratio}\;Cgel=-59.65\frac{XG}{CNC}+25$$
(2)$$\mathrm{At}\;\mathrm{high}\;\mathrm{ratio}\;Cgel=4\frac{XG}{CNC}+4$$

The numerical determination of the intercept point of Equations (1) and (2) led to the following solution:$$\frac{XG}{CNC}=0.33\;leading\;to\;Cgel=5.31\;g\;L^{-1}$$

The XG/CNC ratio of 0.33 is consistent with the limit value for saturation of the CNC surface by XG, which is of the order of 0.25 to 0.3 g XG/g CNC as previously determined [4]. When combining the QCM-D adsorption results with these values, it can be proposed that the ratio XG/CNC = 0.33 corresponds to a mechanical change in the gelation process. At ratios below 0.33, the amount of surface area available is greater than the quantity of XG needed to saturate the surfaces, so XG adsorption takes place preferentially in extended conformation, leading to CNC cross-linking. Above 0.33, the surfaces are saturated, and XG adsorption leads to a loop and tail formation, increasing the hydrodynamic volume of the XG/CNC complexes and leading to the individualization of the XG/CNC complexes. In summary, when the XG amount is lower than the number of sorption sites on the CNC (i.e., CNC surface available), then XG adopts an extended conformation consistent with the QCM-D data. When the XG amount is higher than the number of sorption sites on the CNC (i.e., CNC surface not available), then XG adopts a loop and tail conformation, and since the surface is saturated, cross-linking is unlikely. The crossover between the two conformations is the CNC surface saturation concentration (from 0.25 to 0.3 g XG/g CNC), which corresponds to the ratio for the minimum gelation concentration and the highest mechanical properties.

The impact of the XG/CNC ratio was also assessed by examining the mechanical properties of mixtures at a constant CNC + XG concentration of 20 g L^−1^. Figure 4a shows the evolution of the storage modulus as a function of the XG/CNC ratio, and Figure 4b displays the evolution of the storage (*G*′) and loss moduli (*G*″) at 1 Hz as a function of the XG/CNC ratio. Neat XG in solution at 20 g L^−1^ displays a typical viscous behavior since *G*′ remains lower than *G*″ over the all-frequency range investigated (Appendix A). Upon adding CNC (XG/CNC > 4), the mixtures remain viscous but the frequency dependence of *G*′ is different from the behavior of neat XG (Figure 4a and Appendix A). For the mixture at XG/CNC = 6, at low frequencies, the presence of a plateau for *G*′ can be seen whose magnitude increases with the increasing amount of CNC in the mixtures. At the highest frequencies investigated, the values of *G*′ remain the same as for neat XG despite the addition of CNC. For XG/CNC = 4, *G*′ is of the same order of magnitude as *G*″ (and both of the same scale as ω^0.5^, which is typical of a system displaying a sol-to-gel transition (Appendix A). For XG/CNC ≤ 4, *G*′ has higher values than *G*″, which is typical of a gel behavior. By decreasing the XG/CNC ratio, the frequency dependence of *G*′ decreases, and the value of *G*′ increases steadily up to XG/CNC = 0.25. It is interesting to note that this value of the XG/CNC ratio is in agreement with the value determined in the phase diagram as the XG/CNC ratio corresponding to the minimal gelation concentration and those where the surface of the CNCs is completely covered by the XG (XG/CNC = 0.2–0.3) [4]. If this ratio is further reduced, meaning that more CNC is added to XG, the values of *G*′ decrease but remain almost independent of frequency.

In summary, the addition of CNC to XG induces a sol-to-gel transition for XG/CNC ratios ranging from four to six for a 20 gL^−1^ XG + CNC concentration. Below this value, the further addition of CNC to XG strengthens the composite hydrogels until reaching the full coverage limit of the surface of the CNC (XG/CNC = 0.2–0.3). For a high XG/CNC ratio (XG/CNC = 2–6), CNCs are covered by XG chains, which are in a huge excess with respect to the surface that is available, meaning that free XG chains are likely to be present in solution [4]. As a result, CNCs behave as cross-links within the semi-dilute solution of XG leading to the appearance of a plateau for *G*′ at low frequency whose magnitude increases with the addition of CNC. The number of free XG chains decreases concomitantly with the volume fraction of XG/CNC complexes increasing. When the surface of the CNCs are no more saturated by XG (XG/CNC < 0.2–0.3), there is a default of XG, and bundles of CNC are formed instead, leading to particulate gels. The mechanical properties of the latter decrease, likely in reason of poor cross-linking and a limited increase in hydrodynamic volume. These results are consistent with conclusions from previous works indicating that gelation results from complex interplays between polymeric chains and the nanorods, which induce an increase in their hydrodynamic volume resulting in steric hindrance between the nanoparticle–polymer complexes [11].

### 2.2. Reinforcing Xyloglucan Interactions: Toward Thermoresponsive Hydrogels

An alternative strategy for modulating the functional properties of nanocolloidal hydrogels is to change the interactions between the building blocks of the colloidal network, i.e., nanoparticles or polymers. This can be achieved by either covalent or physical modifications. In the former case, the lifetime of covalent bonds being infinite, a modification leads to irreversible changes, whereas the lifetime of physico-chemical interactions being shorter, the created systems can be reversible, and properties can be adjusted and possibly triggered by external stimuli. In the case of natural polymers, an original and effective strategy is enzymatic modification. Enzymes perform highly specific modifications under mild conditions in aqueous media, with very high efficiency. In the case of XG, several enzymes perform modifications that lead to the modulation of the interactions between chains. Using galactose oxidase, Parikka et al. catalyzed the oxidation of the primary hydroxyl groups of the terminal galactose units of XG from tamarind seeds, leading to cross-linking of these polysaccharides by hemiacetal bonds [25]. XG cross-linking results in the gelation of XG solutions or the enhancement of the mechanical properties of mixtures of nanofibrillated cellulose and XG [26,27]. Similarly, beta-galactosidase undergoes a degalactosylation of XG chains leading to the modulation of their associative properties [28,29]. Indeed, this modification confers a thermosensitive character to XG, since a thermally induced sol–gel transition at 35 °C has been characterized when the galactose content is decreased by around 40% [30]. For degalactosylated XG, thermothickening is attributed to altered water/polymer interactions leading to the aggregation of hydrophobic domains (i.e., the degalactosylated portions of XG chains) to minimize the hydrophobic surface area in contact with bulk water. This property has been reported not only for degalactosylated XG but also for various cellulose derivatives, nanocellulose/cellulose derivative blends, and synthetic polymers [31,32,33]. Furthermore, de Freitas et al. demonstrated that heating pure XG solutions induced the formation of a quasi-permanent network that could be consistent with the results presenting this thermal transition [34]. We recently demonstrated that mixtures between CNC and XG having been degalactosylated (DGXG) could form thermosensitive hydrogels [35]. In this previous study, DGXG was prepared prior to mixing with CNCs, and self-association processes leading to phase separations were highlighted. To overcome this drawback, we investigated the straightforward modification of XG/CNC complexes. To this end, XG/CNC blends at different ratios were formulated and then subjected to β-galactosidase. The evolution of the mechanical properties of the blends is shown in Figure 5 for three XG/CNC ratios (XG/CNC = 0.25, 1.5, and 4).

The action of degalactosylation on the mechanical properties of XG results in an increase in storage and loss moduli between 30 and 40 °C [30,34]. This transition is only visible in the case of the highest XG/CNC ratios (1.5 and 4) whereas in the case of the lowest ratio (0.25), the mixture displays a slight thermothinning behavior since both moduli decrease upon increasing the temperature (Figure 5). This lack of transition at low ratios can be either explained by the limited accessibility of XG when intimately bound in extended form to CNC, which is the case for a ratio of 0.25, or by the lack of mobility of the modified chains that can limit their self-assembly. In the case of higher XG/CNC ratios (1.5 and 4), thermothickening is observed around a value of 35 °C, which is consistent with the transitions already observed on isolated XG or DGXG/CNC complexes [30,34]. This transition is reversible though displaying a very strong hysteresis. After several hours of cooling, the mixtures recover their initial mechanical properties. These results need to be further investigated in subsequent work to fully explore the fine tuning that can be achieved in terms of transition temperature, formation kinetics, hysteresis, or modulation of mechanical properties of the nanocomposite hydrogels.

### 2.3. Crowding Effect

Achieving an XG/CNC hydrogel with a high solid content is difficult because the dispersion of XG at concentrations above 3–4 wt.% results in solutions displaying very high viscosities not to mention the difficulty of hydrating XG at higher concentrations. To extend the range of concentrations studied, hydrogels were concentrated by osmotic dehydration (OD), also known as osmotic compression [36,37]. OD involves immersing hydrogels sealed in a dialysis tube into a concentrated solution of polyethylene glycol. Water then diffuses through the membrane to balance the osmotic pressure with XG and CNC retained in the tube, resulting in an increase in their concentration. In our case, a 5% *wt*/*wt* poly(ethylene glycol) solution was used for generating an osmotic pressure of 3 × 10^4^ Pa [38]. OD has been previously used to dehydrate hydrogels and polymer nanoparticles, including nanocellulose [17,39,40,41]. OD was performed on the CNC dispersion, XG solution, and three selected hydrogels with different ratios (0.25, 1.5, and 4, starting from a concentration of 20 g L^−1^) representing low, medium, and high XG/CNC ratios. The osmotic pressure is a colligative property that counts the number density of contributing species [42]. Because the composition and the nature of the objects (nanoparticles, free XG, XG–CNC complexes) varied in the three samples, various final concentrations were achieved (Figure 6). The final concentration of CNC was 130 g L^−1^ while that of XG reached 80 g L^−1^. For the hydrogels, the highest concentration was achieved with the lowest XG/CNC ratio of 0.25 (165 g L^−1^). These results are consistent with the fact that at this ratio, XG/CNC complexes are cross-linked, and therefore, the number of objects is reduced. Moreover, the XG–CNC complexes are probably already denser due to the low XG amount that closely connects CNC, and therefore, at this ratio, hydrogel is easier to compress. The lowest final concentration was obtained with the XG/CNC ratio of four since in that case, the hydrogel likely consisted of a mixture or XG–CNC complexes and free XG chains so that the number of objects was probably higher. The hydrogel with a XG/CNC ratio of 1.5 was an intermediate case.

For the three composite XG/CNC hydrogels, self-standing gels were obtained while XG and CNC samples still displayed viscous properties despite their high concentration. The mechanical properties of the composite hydrogels were investigated by rheology (Figure 7a,b). OD induced a strong increase in the mechanical properties since *G*′ and *G″* displayed values two orders of magnitude higher than at 20 g L^−1^ (Figure 5). Discussing the influence of the XG/CNC ratio on them is rather difficult since the overall concentration of the samples was not the same. However, the dependence of *G*′ on the frequency became less important upon decreasing the XG/CNC ratio.

Finally, we investigated the hydrogels’ behavior when submitted to large strains (Figure 7b). For all XG/CNC ratios above 0.25, *G*′ and *G*″ remained constant as the strain increased until reaching a critical strain where the systems were disrupted. By further reducing the strain, no hysteresis was measured, which means that the systems recovered their properties very rapidly. For XG/CNS = 0.25, *G*″ passed through a maximum before decreasing when increasing the strain, which accounted for the presence of a yield stress. Such a pattern was evidenced for gels of CNCs in the presence of salt [43] or poly(vinyl alcohol) PVA [44]. In both cases, it was attributed to the formation of physical cross-links promoted under shear through either a decrease in the interparticle distance for a CNCs/salt mixture and bridging of CNCs by PVA chains. Such a bridging promoted by shear is likely to occur at XG/CNS = 0.25, where the surface of the CNCs are close to being saturated. Furthermore, in this peculiar case, the shear-induced bridging could account for the small hysteresis that can be detected for the strain dependence of the moduli upon increasing or decreasing the strain [43]. However, despite this hysteresis, the mechanical properties fully recovered their initial values when the strain was reduced. XG/CNC hydrogels based on XG with a lower molecular weight as well as aggregated CNC in the presence of salt displayed a similar behavior, which was explained by the presence of particles–polymer complexes structured in domains/flocs reorganizing under shear [43]. Actually, the XG/CNC complexes being less aggregated, they needed less strain to reorient under shear. Furthermore, once the strain was released, the mechanical properties exhibited a rapid recovery and no hysteresis. We can take advantage of this property by molding the nanocomposite hydrogels into various shapes. As shown in Figure 7c, the gel can be ground and then pressed into a mold to quickly acquire the desired shape. The same sample can be ground again and remolded into another shape, demonstrating very good self-healing properties.

## 3. Conclusions

Combining CNC and XG by blending offers a simple, environmentally friendly route to manufacturing biobased materials. By varying interactions and formulation parameters, a very broad spectrum of functional properties can be accessed and fine-tuned. In this work, we described a few possibilities offered by these systems by investigating the impact of the nanoparticle/polymer ratio, the modulation of interactions through the use of enzymes, which is a specificity offered by biopolymers, and finally by playing with the solid content in the materials. Depending on the properties being evaluated or targeted, optimum compositions can be identified. For mechanical properties, for example, a ratio of 0.25, which corresponds to the saturation of CNC surfaces by XG, gives the highest mechanical properties both at low (Figure 4) and high concentrations (Figure 7). In other cases, such as the appearance of themosensitive properties, the effective XG/CNC ratio should be bigger than 0.25. For other functionalities, the determination of optimal systems depends on the properties sought. All these elements demonstrate the versatility of the system. This of course means that hydrogels with a wide range of properties can be produced and adjusted, but also that the properties of materials derived from hydrogels, such as aerogels obtained from hydrogels by drying, can be controlled [45]. This question has yet to be studied and is undoubtedly full of promise. XG/CNC systems may then be considered as biobased case studies that can further pave the road of research on being integrated into the class of colloidal nanogels recently described by Kumacheva’s group [12]. Additionally, the three modulation strategies described in this work can be completed by other parameters yet to be described. This include, for example, the synergistic properties of nanoparticles by varying their composition, size, and shape, or taking advantage of the almost infinite structural variability of biopolymers. Biobased nanocolloidal gels should be able to contribute to the emergence of new applications such as soft robotics, smart coatings, optoelectronic devices, biosensors, absorbents, and many others.

## 4. Materials and Methods 

### 4.1. Materials

CNCs (sodium salt form) from CelluForce (Montreal, QC, Canada) were obtained as a spray dried powder. CNCs were produced from bleached Kraft pulp by sulfuric acid hydrolysis. According to CelluForce specifications, CNCs present dimensions of 2.3–4.5 nm in cross-section and 44–108 nm in length (as measured by Atomic Force Microscopy, AFM); a crystalline fraction of 0.88 (measured by X-ray diffraction); and a surface charge density of 0.023 mmol g^−1^ (measured by conductometric titration). Tamarind seed gum XG was purchased from DSP GOYKO FOOD & CHEMICAL (Osaka, Japan) and was used as received. The molar mass and overlap concentration were determined in previous studies and are equal to 8.4 × 10^5^ g mol^−1^ and 0.5 g L^−1^ [35], respectively. Poly(ethylene glycol) of molar mass 35,000 g mol^−1^ was purchased from Sigma Aldrich. Poly(allylamine hydrochloride) (PAH) (*M_w_* = 120,000–200,000 g mol^−1^) was purchased from PolySciences.

CNC suspensions were prepared by dispersing the powder in deionized water (18.2 MΩ cm resistivity, Millipore Milli-Q purification system). The CNC suspensions were sonicated for 5 min at 50% amplitude, with an ultrasonic probe (Q700 sonicator, 20 kHz, QSonica LLC., Newtown, CT, USA) equipped with a 12.5 mm diameter titanium microtip.

XG solutions were prepared by dissolving into deionized water. The solution was gently heated under stirring at 40 °C and left under stirring overnight at 4 °C for complete dissolution.

### 4.2. Quartz Crystal Microbalance with Dissipation (QCM-D)

QCM-D experiments were carried out with a Q-Sense E4 instrument (AB, Gothenburg, Sweden) using a piezoelectric AT-cut quartz crystal coated with gold electrodes on each side (QSX301, Q-Sense). Gold-coated quartz crystals were first immersed in a piranha solution H_2_SO_4_/H_2_O_2_ (7:3, *v/v*), then rinsed and dried under a stream of nitrogen. Prior to use, QCM-D quartz sensors were subjected to a plasma etching device (Harrick Plasma, Ithaca, NY, USA) for 20 min. Surfaces of CNC were prepared by the spin-coating method. A CNC dispersion was dropped on a pre-coated substrate with poly(allylamine hydrochloride) (PAH) at 4 g L^−1^ in water, and after 5 min of adsorption, accelerated at 180 rpm s^−1^ to 3600 rpm for 60 s. Frequency (*Δf_n/n_*) and dissipation changes (*ΔD_n_*) were simultaneously registered at a 5 MHz fundamental resonance frequency, and its several overtones as a function of time. Spin-coated CNC surfaces were placed in the QCM-D cells at 20 ºC and rinsed with water until the resonance response was stable. Then, frequency and dissipation signals were offset to zero just before the measurement. Xyloglucan solutions at different concentrations (0.3–15 µg mL^−1^) were injected at 50 µL min^−1^ and allowed to adsorb for 40 min. Water was then injected in order to remove any loosely bound material. Each XG concentration was adsorbed on a freshly prepared CNC-modified surface and the experiments were repeated at least three times.

### 4.3. Preparation of XG/CNC Mixtures and the Investigation of Their Gelling Properties by the Inverted Test Tube Method

To map a ternary phase diagram, mixtures of XG and CNC were prepared at different weight ratios at room temperature by varying the concentrations of XG (0–20 g L^−1^) and CNC (0–20 g L^−1^) to produce a final dispersion with a volume of 10 mL at a total XG + CNC concentration of 20 g L^−1^. XG and CNC solutions were simultaneously added at 1 mL min^−1^ with a syringe pump. During the addition time, the mixture was thoroughly mixed with a Rotorstator. Once the addition was complete, the mixtures were manually shaken for a few seconds and vortexed before being placed in a cold room at 4 °C. The inverted test tube method was performed for each sample to check whether the mixtures were able to flow under their own weight or not, to discriminate between liquids and gels. Gelling was determined visually by assessing whether or not the mixture was flowing, 5 min after mixing.

### 4.4. Enzymatic Degalactosylation of Xyloglucan

The enzymatic modification of XG was achieved by following the protocol of Brun-Graeppi et al. using *β*-Galactosidase from *Aspergillus oryzae* which was purchased from Sigma-Aldrich (Saint Louis, MI, USA) [30]. Enzymatic modifications were achieved on XG/CNC complexes at 50 °C with an enzyme/substrate ratio of 0.37 U/mg XG. The mixtures were stirred and left to react for 22 h. The mixtures were heated at 90 °C for 5 min to deactivate the enzyme. The mixtures were then cooled and kept at 4 °C.

### 4.5. Rheology

The mechanical properties (storage *G*′ and loss *G*″ moduli) of the gels were measured by using a stress-imposed rheometer AR-2000 (TA Instrument, New Castle, DE, USA) equipped with parallel plates (diameter of 40 mm or 20 mm depending on the strength of the gels). Rheological measurements were performed within the linear response regime of the sample established prior to measurements at a frequency of 1 Hz. All rheological measurements were performed at 20 °C when not specified but when studying the influence of temperature, measurements were performed between 10 °C and 60 °C using heating/cooling rates of 2 °C min^−1^. The samples were covered with paraffin oil to prevent water evaporation.

### 4.6. Osmotic Dehydration

Osmotic compression was used to concentrate the hydrogels. In brief, 40 mL of gel was placed within a dialysis membrane (Spectra/Por 4 Dialysis Membrane WITH MWCO = 12–14 KD) and dialyzed against a solution of PEG at 50 g L^−1^. The mass loss was determined by thermogravimetric analysis.

### 4.7. Molding Assay

The concentrated gels were mechanically crushed into small pieces. The resulting paste was poured into a mold and left at 4 °C overnight before demolding. The following day, the molded gel was crushed a second time and poured into a mold with a different pattern to that used in the first stage, and left to rest again overnight at 4 °C.

## Figures and Tables

**Figure 1 gels-10-00334-f001:**
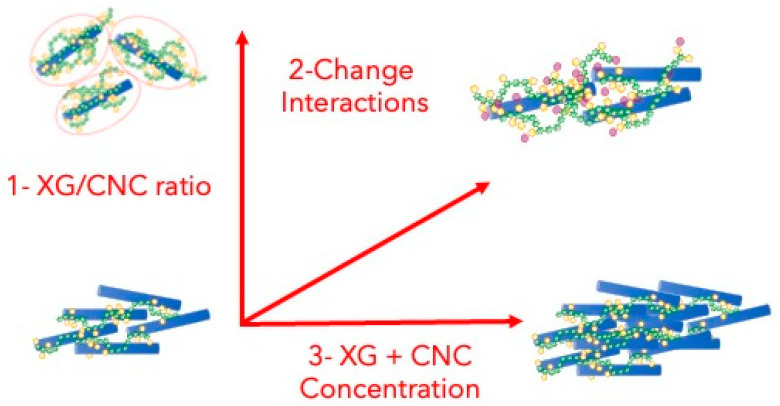
Schematic view of the different levers that were investigated in this paper to tune XG/CNC nanocolloidal hydrogel properties. XG (green pearl necklaces) and CNC (blue rods) are not scaled. The first one corresponds to the variation in the XC/CNC weight ratio, the second one involves the enzymatic modification of the XG structure to change interaction, and finally, the increase in concentration that leads to higher crowding corresponds to the third lever.

**Figure 2 gels-10-00334-f002:**
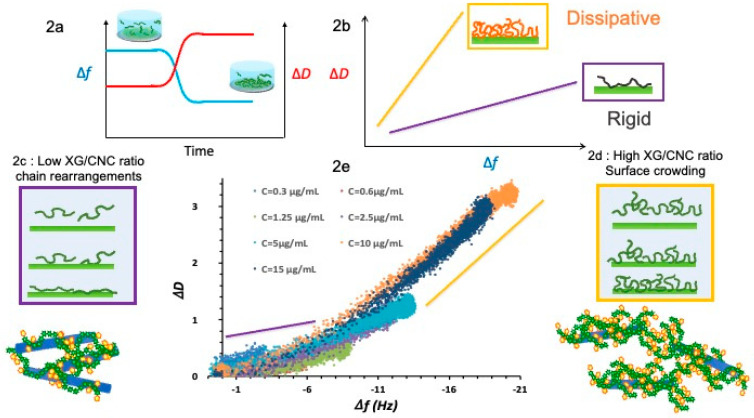
Principle of QCM−D (**2a**) and *ΔD* vs. *Δf* scheme (**2b**). Graph (**2e**) presents the *ΔD* vs. *Δf* plots for different injections of XG/CNC ratio corresponding to XG adsorption at low (**2c**) and high (**2d**) concentration. As the injection concentration increases (from 0.3 μg/mL up to 15 μg/mL), the slopes increase, indicating a more dissipative structure that can be attributed to the formation of loops and tails.

**Figure 3 gels-10-00334-f003:**
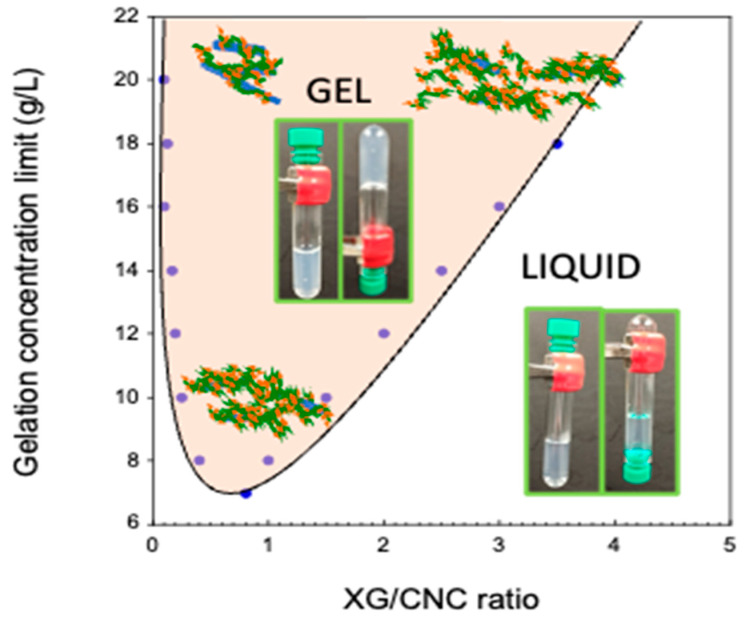
Phase diagram of the XG/CNC system. The gelation concentration (XG + CNC) is plotted against de XG/CNC ratio. The continuous line is merely a guide for the eyes to visualize the gel and liquid states.

**Figure 4 gels-10-00334-f004:**
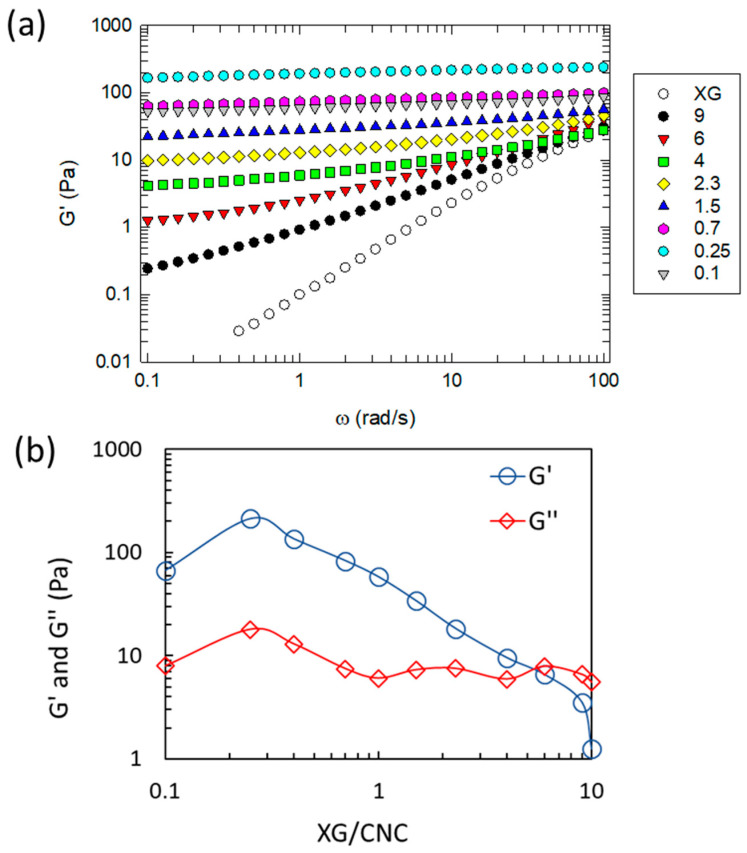
(**a**) Storage modulus as a function of pulsation for neat XG and mixtures at various XG/CNC weight ratios as indicated in the legend of the figure. The total concentration is 20 g L^−1^. (**b**) Values of *G*′ (blue trace) and *G*″ (red trace) obtained for the same hydrogels at 1 Hz as a function of the XG/CNC ratio; lines are a guide for the reader.

**Figure 5 gels-10-00334-f005:**
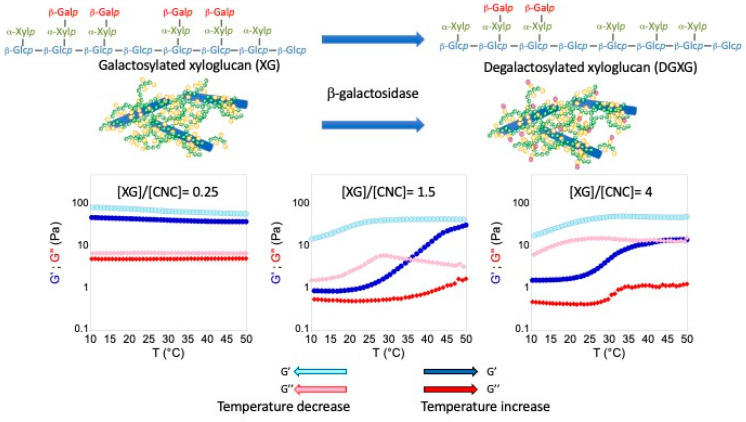
Enzymatic modification of XG by β-galactosidase and temperature dependence of the storage (*G*′) and loss (*G*″) moduli for three different XG/CNC ratios.

**Figure 6 gels-10-00334-f006:**
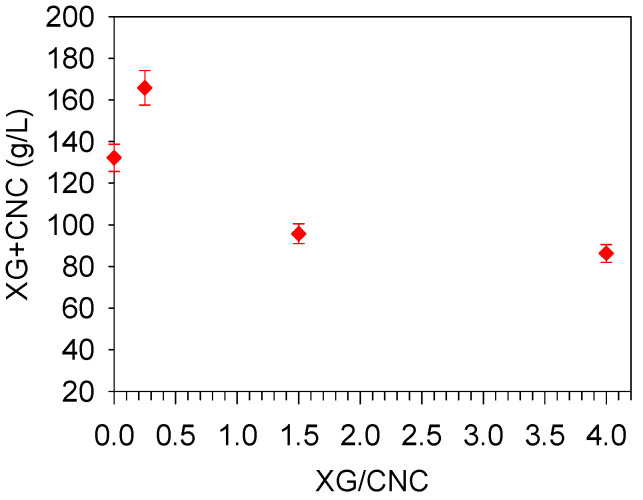
CNC + XG concentration for hydrogels with various XG/CNC ratios obtained after osmotic dehydration starting from a concentration of 20 g L^−1^.

**Figure 7 gels-10-00334-f007:**
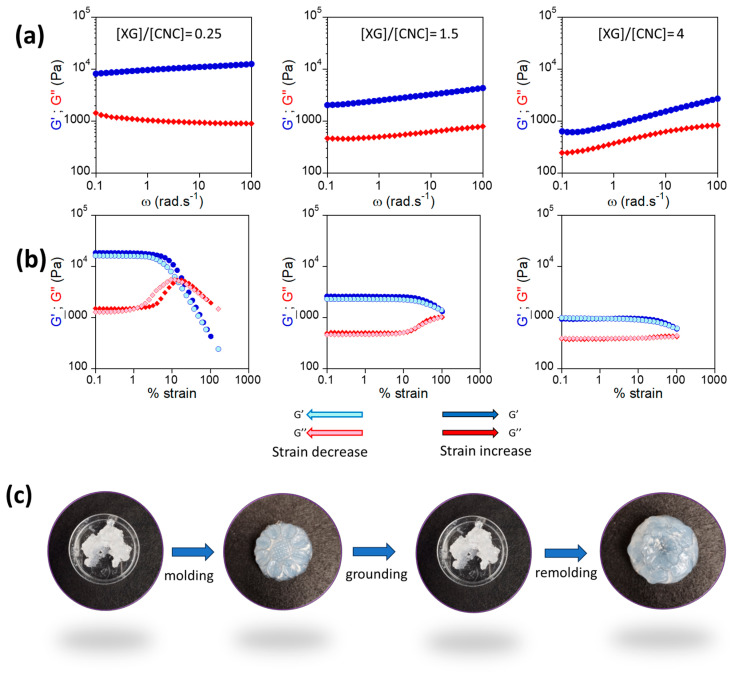
(**a**) Frequency sweeps obtained at 20 °C for storage (blue circle) and loss (red diamond) moduli of hydrogels at various XG/CNC ratios for hydrogels obtained by dehydration; the concentrations were 165, 96, and 86 g L^−1^, respectively. (**b**) Strain sweeps obtained at 20 °C for storage (blue circle) and loss (red diamond) moduli for the same hydrogels. The blue traces correspond to an increase in the strain (dark blue *G*′/dark red *G*″), and the light blue (*G*′)/light red (*G*″) ones to a decrease. (**c**) Macroscopic pictures of composite hydrogels with XG/CNC = 0.25 and C = 165 g L^−1^ before and after being left overnight at 4 °C in a patterned mold.

## Data Availability

All data and materials are available on request from the corresponding author. The data are not publicly available due to ongoing researches using a part of the data.

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
