# Peer review of "Xyloglucan–Cellulose Nanocrystals Mixtures: A Case Study of Nanocolloidal Hydrogels and Levers for Tuning Functional Properties"

_gels, 2024, doi:10.3390/gels10050334_

Round 1

Reviewer 1 Report

Comments and Suggestions for Authors

The manuscript submitted by Rangel et al. to Gels focuses on the preparation and study of the properties of bio-based hydrogels from renewable components, namely nanocrystalline cellulose and xyloglucan. The content of the manuscript is novel and in line with the scope of Gels. The paper is well written, organized and illustrated. In general, the manuscript can be recommended for publication after minor revision. 

1. Page 2, line 83. According to polymer nomenclature, the prefix “poly” refers to the whole name of the monomer. Thus, the name poly(ethylene glycol) should be written in parentheses. 

2. Page 2, line 83. The molecular weight of the polymer should be written as 35,000, not 35.000.

3. Page 6, line 220. Check the concentration value. The dot seems to be placed in the wrong place. 

4. Figures 3 and 5 should be placed after the text where they are mentioned first. 

5. Figure 4, X-axis. Do you mean the mass ratio for XG/CNC?

Reviewer 2 Report

Comments and Suggestions for Authors

The paper is focused on the characterization of hydrogels based on Xyloglucan-cellulose nanocrystals mixtures. The topic falls within the scope of the journal. The presentation and discussion of the results could be partlty improved. On this basis, I recommend the publication after the following revisions:

-          Fig. 5. I suggest to add the dependences of tan(delta) on Temperature. The curves obtained curves should present a peak at the thermal transition. The authors could discuss the temperature peak values by considering the specific Xyloglucan-cellulose nanocrystals ratio. Moreover, it would be interesting to compare these results with that of pristine Xyloglucan.

-          Fig. 6. Please add the error bars.

-          The observation of hysteresis process (detected in Fig. 7b) should be better discussed by considering the specific interactions occurring in the XG/CNC hydrogels with variable composition.    

Comments on the Quality of English Language

Minor corrections are needed. 

Round 2

Reviewer 2 Report

Comments and Suggestions for Authors

The paper was improved according to the reviewers' suggestions. I recommend its publication in the current form. 

Comments on the Quality of English Language

Minor corrections are needed. 
